# Redox Modification of PKA-Cα Differentially Affects Its Substrate Selection

**DOI:** 10.3390/life13091811

**Published:** 2023-08-26

**Authors:** Jeannette Delva-Wiley, Ese S. Ekhator, Laquaundra L. Adams, Supriya Patwardhan, Ming Dong, Robert H. Newman

**Affiliations:** 1Department of Biology, North Carolina A&T State University, Greensboro, NC 27411, USA; jeannette_delva-wiley@med.unc.edu (J.D.-W.); esekhator@aggies.ncat.edu (E.S.E.); llhampton@aggies.ncat.edu (L.L.A.); supriya.p02@gmail.com (S.P.); 2Department of Chemistry, North Carolina A&T State University, Greensboro, NC 27411, USA; 3Department of Chemistry and Biochemistry, University of North Carolina Wilmington, Wilmington, NC 28403, USA; dongm@uncw.edu

**Keywords:** cAMP-dependent protein kinase (PKA), protein kinase, oxidation, reactive oxygen species, redox signaling, phosphorylation-dependent signaling

## Abstract

The cyclic AMP-dependent protein kinase (PKA) plays an essential role in the regulation of many important cellular processes and is dysregulated in several pervasive diseases, including diabetes, cardiovascular disease, and various neurodegenerative disorders. Previous studies suggest that the alpha isoform of the catalytic subunit of PKA (PKA-Cα) is oxidized on C199, both in vitro and in situ. However, the molecular consequences of these modifications on PKA-Cα’s substrate selection remain largely unexplored. C199 is located on the P + 1 loop within PKA-Cα’s active site, suggesting that redox modification may affect its kinase activity. Given the proximity of C199 to the substrate binding pocket, we hypothesized that oxidation could differentially alter PKA-Cα’s activity toward its substrates. To this end, we examined the effects of diamide- and H_2_O_2_-dependent oxidation on PKA-Cα’s activity toward select peptide and protein substrates using a combination of biochemical (i.e., trans-phosphorylation assays and steady-state kinetics analysis) and biophysical (i.e., surface plasmon resonance and fluorescence polarization assays) strategies. These studies suggest that redox modification of PKA-Cα differentially affects its activity toward different substrates. For instance, we found that diamide-mediated oxidation caused a marked decrease in PKA-Cα’s activity toward some substrates (e.g., Kemptide and CREBtide) while having little effect on others (e.g., Crosstide). In contrast, H_2_O_2_-dependent oxidation of PKA-Cα led to an increase in its activity toward each of the substrates at relatively low H_2_O_2_ concentrations, with differential effects at higher peroxide concentrations. Together, these studies offer novel insights into crosstalk between redox- and phosphorylation-dependent signaling pathways mediated by PKA. Likewise, since C199 is highly conserved among AGC kinase family members, they also lay the foundation for future studies designed to elucidate the role of redox-dependent modification of kinase substrate selection in physiological and pathological states.

## 1. Introduction

Protein kinases play a central role in many cellular signaling processes. Though a great deal is known about how kinases function in cellular signal transduction, questions still remain about the mechanisms by which specificity is achieved in cellular signaling processes. For instance, a given kinase may drive one cellular process (e.g., cell proliferation) under one condition while promoting a different set of outcomes under another condition (e.g., apoptosis). While the spatial and temporal regulation of kinase activity is believed to play an important role in determining when and where a given kinase is activated inside the cell, crosstalk with other signaling pathways also plays an important role in achieving signaling specificity. Recently, crosstalk between redox- and phosphorylation-dependent signaling pathways has been implicated in various pervasive diseases, including diabetes, cancer, cardiovascular disease, and several neurological disorders, including Alzheimer’s disease [1,2,3,4,5,6,7,8,9,10,11,12,13,14,15]. Therefore, we sought to gain a better understanding of the mechanisms of crosstalk between phosphorylation- and redox-dependent signaling. Specifically, we examined the impact of both diamide- and hydrogen peroxide (H_2_O_2_)-dependent oxidation on the substrate selection of the canonical serine/threonine (S/T) protein kinase family member, cAMP-dependent protein kinase (PKA; Uniprot ID: P05132).

PKA is one of the most well-characterized protein kinases in the kinome [16,17,18]. In the absence of cAMP, PKA typically exists as a holoenzyme composed of two regulatory subunits (PKA-R; R-subunit), each bound to one catalytic subunit (PKA-C; C-subunit) [18]. The enzymatic activity of PKA-C is inhibited in the context of the holoenzyme through its association with the R-subunits. Though both PKA-R isoforms, PKA-RI and PKA-RII, inhibit PKA-C activity with a K_i_ < 1 nM, the means by which they do so are quite different [19]. For instance, while the PKA-RII isoform inhibits PKA-C activity by preventing dissociation of ADP following phosphorylation of a PKA consensus phosphorylation motif (i.e., [R/K]-[R/K]-x-[S/T]) located within PKA-RII’s recognition sequence, the recognition sequence of PKA-RI contains a pseudosubstrate domain in which the phosphoacceptor site has been mutated to either Ala (in the case of the PKA-RIα splice variant) or a Gly (in the case of PKA-RIβ). Despite these divergent modes of inhibition, both PKA-RI and -RII bind PKA-C with sub-nanomolar affinity (K_D_ = ~0.1 nM), sequestering the C-subunit in the holoenzyme and preventing it from phosphorylating its over 250 cellular substrates (which include many metabolic enzymes, transcription factors, ion channels, and chromosomal proteins) [16,20,21]. The R-subunits are also important for targeting the PKA holoenzyme to different subcellular regions via interactions with scaffold proteins, such as A-kinase anchoring proteins (AKAPs), that can bind different substrates and lead to distinct pools of PKA activity upon elevation of intracellular cAMP levels [22,23,24].

In contrast, when intracellular cAMP levels are high, two molecules of cAMP bind each R-subunit via adjacent cyclic nucleotide-binding domains (CNB1 and CNB2) located in the C-terminus of each R-subunit. The cooperative binding of cAMP to CNB1 and CNB2 induces conformational changes in the R-subunit that disrupt interactions between PKA-C and the recognition sequence [25,26,27]. As a consequence, PKA-C is released from the holoenzyme complex, revealing its active site [25]. Importantly, dissociation of PKA-C from the holoenzyme opens the condensed activation loop structure and allows substrates to interact with and become phosphorylated by PKA-C. For instance, both phospho-threonine 197 (pT197), which facilitates reorganization of the active site cleft, and cysteine 199 (C199), which provides a docking site with the P + 1 hydrophobic residue of canonical PKA substrates, are exposed following dissociation of PKA-C from the holoenzyme [28].

Previous studies have shown that C199 can be oxidized using the chemical oxidizer, diamide [28,29]. Diamide-mediated oxidation leads to the formation of a bulky sulfenylhydrazine adduct that can subsequently be converted to a mixed disulfide in the presence of reduced glutathione (GSH). Both diamide-mediated oxidation and glutathionylation led to a decrease in PKA-C activity toward the model PKA substrate, Kemptide, suggesting that redox modification can affect PKA-C activity [28,29]. Consistent with this notion, when C199 was mutated to alanine, no changes in PKA-C activity toward Kemptide were observed following treatment with either diamide or diamide and GSH [28]. However, diamide is not a physiological oxidizing agent and the formation of the sulfenylhydrazine intermediate may mask oxidation-induced changes in PKA-C activity caused by physiological oxidants, such as H_2_O_2_.

Due to its rapid diffusion and relative stability, H_2_O_2_ is the primary signaling molecule in most redox-dependent signaling pathways [11,30,31,32]. H_2_O_2_-dependent oxidation of reactive Cys residues leads to the formation of a sulfenic acid moiety that changes the size and charge of the modified Cys residue [31,32]. While sulfenylation is readily reversible, further oxidation by H_2_O_2_ leads to the formation of higher order oxoforms, such as sulfinic and sulfonic acid, that are largely irreversible inside cells. To guard against overoxidation, sulfenylated residues can react with GSH to form a mixed disulfide. Glutathione, which is a tripeptide composed of Gly, Cys, and an inverted Glu residue, can subsequently be removed through the action of members of the redoxin superfamily (e.g., thioredoxin and glutaredoxin), resulting in the regeneration of a sulfhydyl group [30,31,32].

Given its proximity to the active site and its role in substrate recognition, we hypothesized that redox modification of C199 on PKA-C could alter its interactions with some substrates while having little effect on others. To explore this hypothesis further, we have examined the impact of redox modification on substrate selection of the alpha isoform of PKA-C (PKA-Cα) using a panel of model peptide and protein substrates. To this end, we used a combination of biochemical and biophysical strategies to better understand the impact that redox modification of PKA-Cα has on its substrate selection. These studies suggest that redox modification of PKA-Cα differentially alters its activity toward model substrates. Together, these studies offer insights into the mechanisms of crosstalk between PKA- and redox-dependent signaling at the level of kinase substrate selection and lay a foundation to better understand the impact of redox modification on PKA activity in physiological and pathological states.

## 2. Materials and Methods

### 2.1. Buffers and Reagents

Sodium fluoride, sodium nitrate, sodium orthovandate, manganese chloride, imidazole, dithiothreitol (DTT), H_2_O_2_, diamide, Tris base, Triton-X-100, Tween-20, bromophenol blue, adenyl-imidodiphosphate (AMP-PNP), bovine serum albumin (BSA), glutathione ethyl ester biotin amide (GSH-biotin), Phusion High-Fidelity polymerase, HisPur Ni-NTA chromatography cartridges, TEMED, PAGE-Ruler pre-stained protein ladder, 1-ethyl-3-(3-dimethylaminopropyl)carbodiimide (EDC), N-hydroxysulfosuccinimde (sulfo-NHS), GeneJet Mini Prep kit, Super Optimal broth with Catabolite repression (SOC), beta-mercaptoethanol (BME), disodium salt dihydrate, ethyleneditetraacetic acid (EDTA), reduced glutathione (GSH), glycerol, glycine, HEPES free acid, magnesium chloride-hexahydrate, and sodium dodecylsulfate (SDS), Zeba spin desalting columns, streptavidin-HRP and West Pico PLUS chemiluminescent substrate were purchased from Thermo Fisher Scientific (Waltham, MA, USA). Catalase from bovine liver, calcium chloride, calpain inhibitor LLnL, MG132, OmniPure agarose, phosphatase inhibitor cocktail 2, phosphatase inhibitor cocktail 3, and SNAP i.d. 2.0 plus kits were purchased from MilliporeSigma (Burlington, MA, USA) while H89 was purchased from Torcis (Bristol, UK). Kinase-Glo assay kits, ADP-Glo assay kits, and ultrapure ATP were purchased from Promega (Madison, WI, USA). Protease inhibitor cocktail without EDTA was purchased from Roche (Basel, Switzerland). 40% acrylamide/bisacrylamide (19:1) solution, 4–15% Tris-Glycine-SDS PAGE gels, and Trans-Blot Turbo plus kits were purchased from Bio-Rad (Hercules, CA, USA). BL21 CodonPlus chemical competent *E. coli* cells and NEB10 chemical competent *E. coli* cells were purchased from New England Biolabs (Ipswich, MA, USA). CREBtide and Kemptide were purchased from Alfa Aesar (Ward Hill, MA, USA). Crosstide, Crosstide(P-3→R), Sure-PAGE Tris/MES/SDS PAGE gels, the eStain L1 protein staining device, the eBlot L1 transfer system and associated products were purchased from Genscript (Piscataway, NJ, USA). IRDye Goat anti-mouse 680 RD, Streptavidin 800CW, goat anti-mouse RD800 were purchased from Li-Cor (Lincoln, NE, USA). HiTrap SP-sepharose cation exchange column was purchased from Cytiva (Marlborough, MA, USA) while anti-PKA-Cα antibody was purchased from BD Biosciences (San Jose, CA, USA).

### 2.2. Plasmids and Site-Directed Mutagenesis

The pET15b-PRKACA plasmid (Addgene plasmid ID: 14921) encoding, from the N- to C-terminus, a hexahistidine tag (H_6_), a thrombin cleavage site, and murine PKA-Cα under the control of the T7 promoter, was a kind gift from Dr. Susan S. Taylor at the University of California at San Diego. Likewise, GST-CREB and deca-histidine-tagged monomeric streptavidin (H_10_-mSA) were purchased from Addgene (plasmids #82721 and #80706, respectively). To mutate C199 in PKA-Cα to either Ser (C199S) or Ala (C199A), a modified QuikChange mutagenesis protocol was employed. Accordingly, forward and reverse primers were designed such that the mutation site overlapped at the 5′-end of the forward primer and the 3′-end of the reverse primer rather than at the center of the primers, as is typical during QuikChange mutagenesis (Table 1). The melting temperature of each primer was 68.4 ± 0.9 °C, with the T_m_’s of the given primer pair differing by no more than 0.9. All primers were synthesized by Eton Bioscience (Research Triangle Park, NC, USA).

To introduce a given mutation, 0.5 mM of the corresponding primer set (Table 1) was incubated together with <250 ng of the pET15b-PRKACA plasmid, 200 mM dNTPs, and 1 unit of Phusion High-Fidelity DNA polymerase using the settings outlined in Table 2. The 5′-ends of the PCR products were then phosphorylated using 10 units of polynucleotide kinase (PNK) before being incubated with 1 μL of T4 DNA ligase for 2 h at RT to ligate the ends and form the pET15b-PRKACA plasmid with the desired mutation. The ligation products were then transformed into chemically competent NEB-10 dh5α *E. coli* and grown overnight at 37 °C on Luria broth (LB)/agarose plates supplemented with 100 mg/mL of ampicillin (Amp). Individual colonies (five colonies per mutant gene) were picked from each LB/Amp plate and used to inoculate 2 mL of LB/Amp culture media. The cultures were grown overnight at 37 °C with vigorous shaking. Finally, plasmids were purified using GeneJet plasmid prep kits, according to the manufacturer’s protocol, and sequenced by Eton Biosciences. Once the mutation-of-interest was confirmed, the corresponding plasmid was transformed into BL21 λDE3 CodonPlus *E. coli* for protein expression.

### 2.3. Bacterial Expression and Purification of Wild-Type PKA-Cα and Variants

Wild-type PKA-Cα and each of the PKA-Cα mutant proteins were expressed in the *E. coli* BL21 λDE3 CodonPlus expression strain and purified by liquid chromatography, as described by Steichen et al. [33]. Briefly, a single colony from an LB/Amp plate was used to inoculate a 10 mL LB/Amp starter culture and grown overnight at 37 °C with vigorous shaking. The following morning, the starter culture was diluted into 500 mL of LB media and grown at 37 °C with vigorous shaking until the optical density at 600 nm (OD_600_) reached 0.8 ODU. Expression of the kinase-of-interest (i.e., WT PKA-Cα, PKA-Cα (C199S), or PKA-Cα C199A) was then induced by the addition of 0.5 mM isopropylthiogalactoside (IPTG) and the culture was incubated at 18 °C for an additional 12–14 h. Following induction, cells were harvested by centrifugation at 3500× *g* for 20 min at 4 °C. The supernatant was discarded, and the pellet was resuspended in 1 mL of ice-cold PBS prior to a second centrifugation step. Finally, after the second centrifugation step, the supernatant was again removed and the pellet was stored at −80 °C until purification. 

To purify H_6_-PKA-Cα and H_6_-PKA-Cα variants, the cell pellet was first resuspended in 30 mL of ice-cold lysis buffer (50 mM potassium phosphate, 20 mM Tris-HCl, pH 8.8, 100 mM NaCl, 50 mg lysozyme, 0.01% (*v/v*) phosphatase inhibitor cocktails 2 and 3, 2 mM NaF, 2 mM Na_3_(VO_4_), 1 mM phenylmethylsulfonyl fluoride (PMSF), and 2 mM β-mercaptoethanol (BME)) by gently pipetting up and down. Cells were then lysed by sonication using the following settings: amplitude = 70%; duty cycle = 1 s on and 1 s off. The resuspension was sonicated using these settings for 1 min followed by a 1 min rest period on ice until the desired sample viscosity was obtained. Following sonication, the cell debris was pelleted by centrifugation at 22,700× *g* at 4 °C for 50 min using a Beckman JM20 rotor. The clarified lysate was then loaded onto a 1 mL HisPur Ni-NTA column that had been pre-equilibrated in lysis buffer lacking protease and phosphatase inhibitors. The flow-through (FT) fraction was collected during each load. After all of the lysates had been loaded, the column was washed with 10 column volumes (CV) of wash buffer 1 (50 mM KH_2_PO_4_, 20 mM Tris-HCl, pH 8.8, 100 mM NaCl, 10 mM Imidazole, 5 mM BME) followed by 10 CV of wash buffer 2 (50 mM KH_2_PO_4_, 20 mM Tris-HCl, pH 8.8, 1 M NaCl, 10 mM Imidazole, 5 mM BME). Immobilized H_6_-PKA-Cα was then eluted using a linear gradient moving from elution buffer A (50 mM KH_2_PO_4_, 20 mM Tris-HCl, pH 88, 100 mM NaCl, 5 mM BME, 50 mM Imidazole) to elution buffer B (50 mM KH_2_PO_4_, 20 mM Tris-HCl, pH 8.8, 100 mM NaCl, 5 mM BME, 500 mM Imidazole) over 10 CV. Elution fractions were analyzed by SDS-PAGE followed by Coomassie staining using either colloidal Coomassie reagent or the eBlot gel staining system. To assess the kinase activity of each fraction, the activity toward Kemptide was measured using the Kinase-Glo activity assay, as described below. Those fractions that contained high levels of the kinase-of-interest and exhibited robust kinase activity were pooled and dialyzed into SP buffer A (20 mM KH_2_PO_4_, pH 6.5, 1 mM DTT, 20 mM KCl) overnight at 4 °C followed by a second exchange for 2–3 h at 4 °C.

Following dialysis, the pooled Ni-NTA product was loaded onto a 1 mL HiTrap sulfopropyl (SP)-sepharose cation exchange column pre-equilibrated in SP buffer A (20 mM KH_2_PO_4_, pH 6.5, 20 mM KCl, 1 mM DTT). Once loaded, the column was washed with SP Buffer A until the A_280_ returned to baseline (typically 10 CV). The protein product was then eluted using a linear KCl concentration gradient ranging from 0.02–1.0 M KCl. As before, the purity of each fraction was assessed by SDS-PAGE followed by Coomassie staining (Appendix A) while the activity of each fraction was analyzed using the Kinase-Glo activity assay toward 50 mM Kemptide. Fractions containing high levels of purified H_6_-PKA-Cα are then pooled and dialyzed overnight at 4°C into storage buffer (20 mM KH_2_PO_4_, 1 mM DTT, 20 mM KCl, and 10% glycerol) followed by a second exchange for an additional 2–3 h at 4 °C using fresh storage buffer. Dialyzed samples were aliquoted into 10–20 mL aliquots and stored at −80 °C until use.

### 2.4. Bacterial Expression and Purification of GST-CREB

GST-CREB proteins were expressed in the *E. coli* BL21 λDE3 CodonPlus expression strain and purified by liquid chromatography, as described below. After transformation, a single colony from the LB/Amp plate was used to inoculate a starter culture of 10 mL of LB/Amp and grown overnight at 37 °C with vigorous shaking. The following morning, 9 mL of the starter culture was diluted into 500 mL of LB media and grown at 37 °C with vigorous shaking until the optical density at 600 nm (OD_600_) reached 0.8 OD. Next, GST-CREB was induced by adding 0.5 mM IPTG and incubating at 16 °C for 14 h. Following induction, cells were harvested by centrifugation at 3500× *g* for 20 min at 4 °C, and the supernatant was discarded. Next, the pellet was resuspended in 1 mL of ice-cold PBS before the second centrifugation. Finally, the supernatant was discarded after the second centrifugation step, and the pellet was stored at −80 °C until purification. To purify GST-CREB, the cell pellet was first resuspended in 30 mL of ice-cold lysis buffer (54 mM Tris-HCl, pH 7.4, 160 mM NaCl, 1 mM EDTA, 10 mM MgSO_4_, 13 mM MgCl_2_, 0.25 mg/mL lysozyme, 1 mM BME, protease inhibitor cocktail) by gently pipetting up and down. Cells were then lysed by sonication using the following settings: amplitude = 70%; duty cycle = 1 s on and 1 s off. The cells were sonicated using these settings for 1 min, followed by a 1 min rest period on ice until the desired sample viscosity was obtained. Following sonication, the cell debris was pelleted by centrifugation at 22,700× *g* at 4 °C for 50 min using a Beckman JM20 rotor. Next, the clarified lysate was loaded onto a 1 mL GSTrap column (ThermoFisher) pre-equilibrated in lysis buffer lacking protease inhibitor cocktail and lysozyme. The flow-through (FT) fraction was collected during each load. Once the lysate was loaded, the column was washed at 1 mL/min with 20–25 mL of Buffer A (25 mM sodium phosphate, pH 7.2, 150 mM NaCl), and protein was eluted in 1 mL fractions with glutathione elution buffer (6 mg/mL glutathione in Buffer A plus 2 mM DTT). SDS-PAGE was then used to analyze the purity of eluted fractions, followed by Coomassie staining using the eBlot gel staining system (Genscript). Following SDS-PAGE, fractions containing high levels of purified GST-CREB were pooled and dialyzed overnight at 4 °C into storage buffer (25 mM sodium phosphate, pH 7.2, 150 mM NaCl, 1 mM DTT, and 10% glycerol), followed by a second exchange for an additional 2–3 h at 4 °C using fresh storage buffer. Finally, dialyzed samples were aliquoted into 10–20 mL aliquots and stored at −80 °C until use.

### 2.5. Bacterial Expression and Purification of Monomeric Streptavidin

Deca-histidine tagged monomeric streptavidin (H_10_-mSA) was expressed in the *E. coli* BL21 λDE3 Codon Plus expression strain and purified using Ni-NTA immobilized metal affinity chromatography (IMAC). After transformation, a single colony from the LB/Kan plate was used to inoculate a starter culture of 10 mL of LB/Kan and grown overnight at 37 °C with vigorous shaking. The following morning, 9 mL of the starter culture was diluted into 500 mL of LB media and grown at 37 °C with vigorous shaking until the OD_600_ reached 0.8 OD. Next, H_10_-mSA was induced by adding 0.5 mM IPTG and incubating at 16 °C for 14 h. Following induction, cells were harvested by centrifugation at 3500× *g* for 20 min at 4 °C, and the supernatant was discarded. Next, the pellet was resuspended in 1 mL of ice-cold PBS before the second centrifugation. Finally, the supernatant was discarded after the second centrifugation step, and the pellet was stored at −80 °C until purification. To purify H_10_-mSA, the cell pellet was first resuspended in 40 mL of ice-cold lysis buffer (50 mM potassium phosphate, 20 mM Tris-HCl, pH 8.0, 100 mM NaCl, 10 mM Imidazole, 0.25 mg/mL lysozyme, 1 pellet protease inhibitor cocktail, 1 mM phenylmethylsulfonyl fluoride (PMSF), and 5 mM BME by gently pipetting up and down. Cells were then lysed by sonication using the following settings: amplitude = 70%; duty cycle = 1 s on and 1 s off. The cells were sonicated using these settings for 1 min, followed by a 1 min rest period on ice until the desired sample viscosity was obtained. Following sonication, the cell debris was pelleted by centrifugation at (22,700× *g*) at 4 °C for 50 min using a Beckman JM20 rotor. The clarified lysate was loaded onto a 1 mL HisPur Ni-NTA column (ThermoFisher) pre-equilibrated in lysis buffer lacking protease and phosphatase inhibitors. The flow-through (FT) fraction was collected during each load. Once the lysate was loaded, the column was then washed with 10 column volumes (CV) of wash buffer 1 (50 mM KH_2_PO_4_, 20 mM Tris-HCl, pH 7.0, 100 mM NaCl, 10 mM Imidazole, 5 mM BME) followed by 10 CV of wash buffer 2 (50 mM KH_2_PO_4_, 20 mM Tris-HCl, pH 7.0, 1 M NaCl, 20 mM Imidazole, 5 mM BME). Bound H_10_-mSA was then eluted using a linear gradient moving from elution buffer A (50 mM KH_2_PO_4_, 20 mM Tris-HCl, pH 7.0, 100 mM NaCl, 5 mM BME, 50 mM Imidazole) to elution buffer B (50 mM KH_2_PO_4_, 20 mM Tris-HCl, pH 7.0, 100 mM NaCl, 5 mM BME, 500 mM Imidazole) over 10 CV. SDS-PAGE was then used to analyze the purity of elution fractions, followed by Coomassie staining using the eStain gel staining system (Genscript). Those fractions containing high protein-of-interest levels were pooled and dialyzed into SP buffer A (20 mM KH_2_PO_4_, pH 6.5, 1 mM DTT, 20 mM KCl) overnight at 4 °C followed by a second exchange for 2–3 h at 4 °C. Following dialysis, the pooled Ni-NTA product was loaded onto a 1 mL HiTrap sulfopropyl (SP)-sepharose cation exchange column pre-equilibrated in SP buffer A (20 mM KH_2_PO_4_, pH 6.5, 20 mM KCl, 1 mM DTT). Once loaded, the column was washed with SP Buffer A until the A_280_ returned to baseline (typically 10 CV). The protein product was then eluted using a linear KCl concentration gradient ranging from 0.02–1.0 M KCl. The purity of each fraction was assessed by SDS-PAGE, followed by Coomassie staining, and the absorbance was taken. Fractions containing high levels of purified H_10_-mSA are then pooled and dialyzed overnight at 4 °C into a storage buffer (5 mM Na_2_HPO_4_, NaH_2_PO_4_, pH 6.8, 150 mM NaCl, 1 mM DTT, and 10% glycerol) followed by a second exchange for an additional 2–3 h at 4 °C using fresh storage buffer. Finally, dialyzed samples were aliquoted into 10–20 mL aliquots and stored at −4 °C until use.

### 2.6. Redox Modification of Wild-Type PKA-Cα and Variants

To better understand the effects that redox modification of PKA-Cα has on its substrate selection, wild-type (WT) H_6_-PKA-Cα and the mutant proteins were treated with oxidizing agents, as described below.

#### 2.6.1. Redox Modification of PKA-Cα Using Diamide and GSH

To assess the impact of diamide-mediated oxidation on PKA-Cα activity toward various substrates, 0.3 µM of the kinase-of-interest (i.e., either WT PKA-Cα, PKA-Cα (C199S) or PKA-Cα (C199A)) was treated with 0–1.6 mM of diamide for 10 min at room temperature. Where indicated, diamide-treated samples were subsequently incubated with 0–2 mM GSH for 10 min at room temperature. To ensure that the molar ratio of diamide:GSH remained constant, a 1.25-fold molar excess of GSH over diamide was used for all GSH-treated samples. Following redox modification with diamide alone or diamide plus GSH, 2.5 μL of the treated kinase was added to 7.5 μL of reaction mix containing the indicated substrate (i.e., 50 μM Kemptide, 75 μM CREBtide, 75 μM Crosstide, or 75 μM Crosstide(P-3→R)) and incubated at room temperature for 30 min. The extent of phosphorylation was measured by the Kinase-Glo coupled luciferase assay (Promega), as described below.

#### 2.6.2. Redox Modification of PKA-Cα Using H_2_O_2_

To examine the impact of H_2_O_2_-dependent oxidation of PKA-Cα on its substrate selection, the activity of the kinase-of-interest (i.e., either WT PKA-Cα, PKA-Cα(C199S) or PKA-Cα(C199A)) toward 50 μM Kemptide, 75 μM CREBtide, or 75 μM Crosstide was determined following PKA-Cα oxidation by various concentrations of H_2_O_2_. To this end, the kinase-of-interest was desalted using a Zeba spin column before being diluted to a final concentration of 0.25 µM in TBS and incubated with the indicated concentration of H_2_O_2_ (ranging from 0–40 µM H_2_O_2_) for 10 min at room temperature. An untreated control, in which the kinase-of-interest was treated with dH_2_O instead of H_2_O_2_, and a negative control, in which the kinase was replaced with TBS, was included in every set of reactions. Following the H_2_O_2_ treatment, excess H_2_O_2_ was removed by incubation with 1 unit (U) of catalase for 1 min at room temperature. Where indicated, the sample was subsequently incubated with a 5-fold molar excess of GSH-biotin at room temperature for 10 min to promote glutathionylation. Glutathionylation was detected by western blot analysis using fluorescently labeled Streptavidin (see below). Finally, after catalase treatment, 2.5 mL of the kinase of interest was added to 7.5 mL of reaction mix containing the indicated substrate and incubated at room temperature for 30 min. The extent of phosphorylation was measured using the Kinase-Glo coupled luciferase assay, as described below. All reactions were run in either duplicate or triplicate.

### 2.7. Kinase-Glo Activity Assay

The activity of the kinase-of-interest (i.e., either WT PKA-Cα, PKA-Cα (C199S) or PKA-Cα (C199A)) following redox modification was measured using the Kinase-Glo coupled luciferase assay according to the manufacturer’s instructions. Briefly, 2.5 µL of either treated or untreated PKA-Cα at a concentration of 0.25 µM was diluted in 7.5 µL of reaction buffer containing the appropriate concentration of substrate. Unless otherwise indicated, the final concentrations of the substrates were: 50 µM Kemptide, 75 µM CREBtide, 75 µM Crosstide, and 75 µM Crosstide(P-3→R). A negative control sample in which the kinase was replaced with TBS was included in all assays. The solutions were then incubated at room temperature for 30 min. Following incubation, the reaction mix was transferred to a white-walled 96-well plate containing 10 µL of dH_2_O and 20 µL of kinase detection reagent. The solution was incubated for an additional 10 min before measuring the luminescence using an Infinite M200 Pro multimode microplate reader (Tecan, Morrisville, NC, USA). In the presence of molecular oxygen, the Kinase-Glo luciferase in the kinase detection reagent couples the hydrolysis of ATP to the conversion of luciferin to oxyluciferin. The conversion of each molecule of luciferin to oxyluciferin is accompanied by the release of a photon of light, which is measured as luminescence. Because kinase-mediated phosphorylation of the substrate depletes the ATP concentration during the 30 min incubation period, higher kinase activity leads to lower luminescence. Therefore, to convert the measured luminescence signal to kinase activity, the luminescence measured in the negative control sample was divided by the luminescence measured for each experimental sample to yield the normalized kinase activity. The normalized kinase activity of each sample was then divided by that of the untreated control to yield the relative normalized activity.

### 2.8. Steady-State Kinetics Analysis

During steady-state kinetic analysis, we mirrored the conditions used during the titration experiments. Briefly, 250 nM of desalted WT PKA-Cα was pre-incubated at room temperature with either dH_2_O (untreated) or 5 µM of H_2_O_2_ for 10 min before excess H_2_O_2_ was removed by treatment with 1 U of catalase for 1 min. The treated kinase was then diluted in reaction buffer containing the indicated substrate concentrations and incubated for 30 min at room temperature. At the end of the experiment, the kinase activity was quenched by adding the PKA-specific inhibitor, H89, and the activity was measured using the ADP-Glo luminescence assay, as described above. Under these conditions, control experiments verified that <10% of the total substrate was consumed by the reaction, suggesting that the steady-state assumption was valid. The initial velocity (v_0_) was then plotted versus substrate concentration, and the kinetic parameters, V_max_ and K_m_, were determined by non-linear regression analysis using the Michaelis–Menten model in GraphPad Prism 7.0 (San Diego, CA, USA). The apparent catalytic rate constant, *k_cat_*_,*app*_, was determined by dividing V_max_ by the enzyme concentration during the reaction.

### 2.9. Surface Plasmon Resonance Binding Assay

To better understand the impact of H_2_O_2_-dependent oxidation of PKA-Cα on its interaction with CREBtide, we used an OpenSPR-XT system (Nicoya, Inc., Kitchener, ON, Canada) to conduct surface plasmon resonance (SPR) binding experiments under both equilibrium (i.e., equilibrium binding) and non-equilibrium (i.e., kinetics analysis) conditions. To this end, Carboxyl Sensors (Nicoya, Inc.) were first pre-conditioned with 10 mM HCl, pH 2.0 at a flow rate of 150 µL/min, followed by activation using a solution composed of 0.4 M EDC and 0.1 M sulfo-NHS at a flow rate of 20 µL/min. Following activation, 34 ng/µL of CREBtide in immobilization buffer (1X HBS-P+, 10 mM MgCl_2_, 0.01% NP-40, and 10 µM AMP-PNP, pH 7.4) was applied to channel 2 at a flow rate of 10 µL/min until the response units (RU) reached 200–350 RU. Once sufficient CREBtide was immobilized, 310 ng/µL monomeric streptavidin (mSA) was applied at a flow rate of 10 µL/min to both channels 1 and 2 followed by 1 M ethanolamine to bind any remaining sites. During equilibrium binding experiments, 10 µM of PKA-Cα was desalted using a Zeba spin column before being treated with either dH_2_O or 200 µM H_2_O_2_ (corresponding to a 20-fold molar excess of H_2_O_2_ over PKA-Cα) for 10 min at room temperature. Excess H_2_O_2_ was then scavenged using 1 U catalase for 1 min at room temperature before diluting PKA-Cα to the indicated concentrations (ranging from 0.078 µM–10 µM). To allow PKA-Cα to associate with immobilized CREBtide, 100 µL of PKA-Cα at the indicated concentration was applied to the sensor surface in running buffer (1X HBS-P+, 10 mM MgCl_2_, 0.01% NP-40, 0.5% BSA, 10 µM AMP-PNP, pH 7.4) at a flow rate of 20 µL/min. Following association, running buffer alone was applied to the sensor at a flow rate of 20 µL/min for 10 min to allow bound PKA-Cα to dissociate from immobilized CREBtide before the next injection. Finally, at the end of each set of injections, 2M NaCl regeneration buffer was injected over the sensor surface at a flow rate of 20 µL/min to remove any undissociated PKA-Cα prior to the next set of injections. Similarly, during kinetics experiments, the indicated concentrations of either untreated or oxidized PKA-Cα (100 µL) were applied to the sensor surface at a flow rate of 75 µL/min. At this elevated flow rate, mass transfer effects are minimized, allowing for the apparent rates of association (*k_on_*_,*app*_) and dissociation (*k_off_*_,*app*_) to be determined. During each set of SPR experiments (both equilibrium binding and kinetics experiments), each set of injections was repeated in duplicate. Following SPR experiments, sensograms were analyzed using either the equilibrium binding or kinetics analysis modules in the TraceDrawer software package (version 4.3; Ridgeview Instruments, Uppsala, Sweden).

### 2.10. Non-Reducing SDS-PAGE

To determine whether WT PKA-Cα forms intramolecular and/or intermolecular disulfide bonds under our experimental conditions, desalted PKA-Cα was treated with the indicated concentrations of diamide or H_2_O_2_ for 10 min at room temperature. In the case of H_2_O_2_-dependent oxidation, excess H_2_O_2_ was then removed by incubation with 1 U of catalase for 1 min before the reaction was quenched by the addition of 6X SDS loading buffer containing either 25 mM DTT (i.e., reducing) or dH_2_O (i.e., non-reducing). Samples were then incubated at 95 °C for 5 min before being resolved on a 4–15% Tris-Glycine SDS-PAGE gel. Following electrophoresis, gels were rinsed with dH_2_O, transferred to a nitrocellulose membrane using the Trans-Blot Turbo semi-dry transfer system (Bio-Rad), and subjected to western blot analysis using an anti-PKA-Cα antibody, as described below.

### 2.11. Western Blot Analysis

Western blot analysis was used to confirm that PKA-Cα was glutathionylated under our experimental conditions and to monitor PKA-Cα during non-reducing PAGE. To this end, samples were resolved by SDS-PAGE before being transferred to a nitrocellulose membrane using either a Trans-Blot Turbo semi-dry transfer system or an eBlot wet transfer system. During electrophoresis, the samples were run either in the presence of 25 mM DTT (during reducing PAGE) or in the absence of a reducing agent (during non-reducing PAGE and the glutathionylation reactions, so that the putative mixed disulfide bond between PKA-Cα and glutathione-biotin would be preserved). After transfer, the nitrocellulose membrane was rinsed quickly in 15 mL of TBST before being blocked for 1 h at RT in 5% fat-free milk dissolved in Tris-buffered saline supplemented with 0.05% Tween-20 (TBS-T). After blocking, the membrane was rinsed with 20 mL of TBS-T before being placed in the appropriate primary antibody solution overnight at 4 °C. Antibody selection was based on the specific epitope that was being targeted. For instance, to assess glutathionylation, the blot was incubated for 1 h at room temperature with the Streptavidin CW 800 IRE Dye (Li-Cor) diluted 1:10,000 in TBS-T. The blot was then rinsed with TBS-T a total of three times for 15 min each and immediately imaged using a Li-Cor Odyssey FC gel imaging system. Similarly, to detect total PKA-Cα, the blot was incubated overnight at 4 °C with mouse anti-PKA-Cα antibody diluted 1:1000 in TBS-T, 1% BSA. The next morning, the blot was washed three times with TBS-T for 10 min each. The blot was then incubated with goat anti-mouse-IRDye 680 RD secondary antibody (Li-Cor) diluted 1:10,000 in TBS-T, 5% milk for 1 h at room temperature. Finally, the membrane was washed three times with TBST for 15 min each before being imaged using a Li-Cor Odyssey FC gel imaging system. Signal intensity was measured using the Empiria Studio suite (Li-Cor, Lincoln, NE, USA). The signal from GST-biotin was normalized to the PKA-Cα signal intensity.

### 2.12. Fluorescence Polarization Binding Assays

Fluorescence polarization (FP) binding assays were used to test the binding affinity of oxidized and reduced PKA-Cα for FITC-labeled PKI (5–24). Briefly, 25 µL of treated or untreated PKA-Cα at 2× the indicated concentration (ranging from 2 pM to 1 µM of PKA-Cα) was added to 25 µL of 0.5 nM FITC-labeled PKI (5–24) (FITC-PKI) in 2× FP buffer (40 mM MOPS, pH 7.0, 300 mM NaCl, 0.01% CHAPS, 20 mM MgCl_2_, and 0.2 mM AMP-PNP) inside of a 384-well, black nonbinding polystyrene plate. All samples were then incubated at room temperature for 30 min prior to measuring the polarized fluorescence signal for FITC (λ_ex_ = 480 nm, λ_em_ = 535 nm) using the fluorescence polarization module on the Infinite F500 Pro multimode microplate reader (Tecan, Morrisville, NC, USA). All sets were run in duplicate with various controls, consisting of either buffer alone (i.e., no FITC-PKI), 0.5 nM FITC-PKI alone, and FITC alone. All data were evaluated by non-linear regression analysis using the EC_50_ model in GraphPad Prism 7.0 (San Diego, CA, USA).

## 3. Results

### 3.1. Diamide-Mediated Oxidation of PKA-Cα, as Well as Subsequent Glutathionylation, Differentially Affects Its Activity toward Different Substrates

Previously, Humphries et al. demonstrated that PKA-Cα^C199^ is oxidized by diamide and that this modification decreased its activity toward the model peptide substrate, Kemptide [28]. To determine if diamide-dependent oxidation of PKA-Cα has a similar effect on other PKA-Cα substrates, we examined the activity of murine PKA-Cα toward three model PKA substrates, namely Kemptide, CREBtide, and Crosstide. While Kemptide (LRRASLG) and CREBtide (KRREILSRRPSYR) both contain the canonical [R/K]-[R/K]-X-[S/T]-X consensus phosphorylation motif recognized by PKA-Cα, Crosstide (GRPRTSSFAEG) contains a looser PKA consensus motif. Consistent with previous studies, pre-treatment of PKA-Cα with low (i.e., sub-micromolar) concentrations of diamide caused a modest decline in its activity toward Kemptide followed by a more pronounced decrease at higher concentrations of oxidant (Figure 1A). A similar change in activity was also observed if CREBtide was used as the substrate. In contrast, diamide treatment led to a much more gradual decrease in PKA-Cα activity toward Crosstide, reaching only 20% inhibition at the highest diamide concentration tested (versus ~68% and 75% inhibition toward Kemptide and CREBtide, respectively). Similar trends were observed if the diamide incubation step was extended to 20 min (Appendix A). All diamide-dependent changes in activity were abolished when PKA-Cα (C199S) (Figure 1A, dashed lines) was substituted for the wild-type enzyme. Together, these data suggest that diamide-dependent oxidation of PKA-Cα on C199 may alter its ability to interact with its substrates rather than strictly inhibiting its catalytic activity.

Also consistent with previous studies, incubation of PKA-Cα with a 1.25-fold molar excess of GSH following diamide treatment caused a steeper drop in its activity toward Kemptide at low concentrations of diamide (Figure 1B) [28]. The same was true for CREBtide and, to a lesser degree, Crosstide. As expected, no change was observed when PKA-Cα(C199S) was substituted for wild-type enzyme (Figure 1B). Interestingly, as opposed to diamide alone, glutathionylation-dependent inhibition appeared to plateau at concentrations of GSH above 31.25 μM (corresponding to 25 μM diamide). In fact, at these elevated concentrations, glutathionylation appeared to restore some PKA-Cα activity compared to diamide alone. For instance, following treatment with 400 μM diamide, PKA-Cα activity toward CREBtide decreased to 25% of the untreated control sample. However, if the PKA-Cα was treated with 400 μM diamide and subsequently incubated with 500 μM GSH, its activity only decreased to 55% of the untreated control. As a result, the difference in activity between samples treated with diamide plus GSH and those treated with diamide alone was characterized by a biphasic profile, with a valley at ~25 μM diamide (and 31.25 μM GSH) followed by an upward trajectory thereafter (Figure 1C). As before, this trend was most pronounced for Kemptide and CREBtide, with Crosstide showing a much smaller overall change in activity between the two conditions.

To better understand the basis for these observations, we used non-reducing PAGE to examine the ultrastructure of PKA-Cα following diamide-mediated oxidation and subsequent glutathionylation (Figure 1D and Appendix A). These experiments suggest that diamide-mediated oxidation of PKA-Cα leads to the formation of both intramolecular disulfide bonds and higher-order oligomers caused by intermolecular disulfide bonds (presumably due to disulfide bonds between C199 and/or C343 residues on separate PKA-Cα molecules). Subsequent treatment with a 1.25-fold molar excess of GSH eliminated the intramolecular disulfide bonds (Appendix A) and decreased the extent of intermolecular disulfide bonding approximately two-fold (Appendix A). Interestingly, simultaneous incubation of PKA-Cα with diamide and GSH also prevented the formation of intramolecular disulfide bonds and virtually eliminated higher-order structures caused by intermolecular disulfide bonding.

### 3.2. H_2_O_2_-Dependent Oxidation of PKA-Cα Alters Its Activity toward Different Substrates Differently

Together, our data suggested that diamide-mediated oxidation of PKA-*Cα* differentially alters its activity toward different substrates, presumably due to the formation of both intramolecular and intermolecular disulfide bonds that may alter substrate binding. However, diamide is a chemical oxidizer that is not typically found under physiological conditions. Therefore, to explore the impact of redox modification of PKA-Cα using a physiological oxidizing agent, we treated PKA-C*α* with various concentrations of H_2_O_2_ and, after scavenging excess H_2_O_2_ with catalase, measured its activity toward Kemptide, CREBtide, and Crosstide (Figure 2A). Interestingly, pre-treatment with low concentrations of H_2_O_2_ led to a dose-dependent increase in PKA activity toward all three substrates. In each case, the activity peaked at 5 µM H_2_O_2_ (corresponding to a 20-fold molar excess of H_2_O_2_ over PKA-C*α*) followed by differential changes in activity toward each substrate at higher H_2_O_2_ concentrations. Strikingly, in contrast to diamide, PKA-C*α* does not appear to form either intra- or intermolecular disulfide bonds under these conditions, as evidenced by non-reducing PAGE (Figure 2B). As expected, mutation of C199 to either Ser or Ala completely abolished H_2_O_2_-dependent changes in PKA-C*α* activity toward the substrates (Figure 2A, dashed lines). Moreover, control experiments using biotinylated GSH demonstrated that, following treatment with 5 μM H_2_O_2_, PKA-C*α* could be glutathionylated using a 5-fold molar excess of GSH, suggesting that PKA-C*α* is sulfenylated under our conditions (Figure 2C). Molecular dynamics simulations using ChimeraX suggest that sulfenylated C199 (C199-SOH) is well accommodated in the PKA-C*α* active site (Appendix A) [34].

To better understand the impact of H_2_O_2_-dependent oxidation on PKA-Cα-substrate interactions, we next employed a series of complementary biophysical and biochemical strategies. For instance, surface plasmon resonance (SPR) is a versatile, label-free method that utilizes ligand-dependent changes in the refractive index at the surface of a gold-plated sensor surface to measure biophysical parameters, such as the binding affinity and association/dissociation kinetics characteristic of protein-protein interactions, in real-time [35]. SPR-based analysis of PKA-C*α*-CREBtide interactions suggests that H_2_O_2_-dependent oxidation of PKA-C*α* decreases its affinity for CREBtide (Figure 3A,B). For instance, under equilibrium conditions, untreated PKA-C*α* bound CREBtide with an apparent dissociation constant (K_D,app_) of 3.3 ± 1.0 µM, which is very similar to the reported K_D_ of 3.9 µM for the PKA-C*α*-CREBtide interaction [36]. In contrast, pre-treatment of PKA-Cα with a 20-fold molar excess of H_2_O_2_ increased its K_D,app_ to 7.2 ± 0.8 µM, corresponding to a 2.2-fold decrease in affinity. Kinetics analyses using SPR suggest that the altered affinity of PKA-C*α* for CREBtide is due primarily to changes in the apparent off-rate (*k*_off,app_) (Figure 3C and Table 3). Indeed, while no significant changes in *k*_on,app_ were observed following H_2_O_2_-dependent oxidation of PKA-C*α* (6.71 ± 1.0 × 10^3^ M^−1^s^−1^ in the reduced state vs. 8.1 ± 1.6 × 10^3^ M^−1^s^−1^ in the oxidized state; *p* = 0.53), *k*_off,app_ increased over 2.9-fold following H_2_O_2_ treatment (1.6 ± 0.25 × 10^−2^ s^−1^ in the reduced state vs. 4.8 ± 1.6 × 10^−2^ s^−1^ in the oxidized state; *p* = 0.015). This yielded a 2.2-fold change in K_D,app_, which is very similar to what was observed during the equilibrium binding experiments. Consistently, similar oxidation-induced changes in the affinity of PKA-C*α* for full-length CREB fused to an N-terminal glutathione-S-transferase tag (GST-CREB) were also observed when binding was probed using steady-state kinetics analysis (Figure 4A and Table 4). For instance, under reducing conditions, PKA bound GST-CREB with a K_m,app_ of 3.4 ± 1.9 µM, which is very similar to the K_D,app_ observed for CREBtide under equilibrium binding conditions using SPR. Meanwhile, pre-treatment with a 20-fold molar excess of H_2_O_2_ increased PKA-C*α*’s K_m,app_ to 29.8 ± 12.4 µM, corresponding to an ~8.8-fold decrease in affinity. It is interesting to note that, though PKA-C*α*’s affinity for both CREBtide and GST-CREB decreased following H_2_O_2_-dependent oxidation of PKA-C*α*, the magnitude of the change was increased substantially when the full-length protein was used, perhaps due to a greater degree of steric hindrance in the full-length protein compared to the smaller peptide. Nonetheless, together, these data suggest that H_2_O_2_-dependent oxidation of PKA-C*α* decreases its affinity for CREB/CREBtide.

Unfortunately, due to the relatively weak interactions between PKA-C*α* and the other peptides (i.e., Kemptide and Crosstide), we were not able to accurately measure PKA-C*α*’s affinity for these substrates using SPR. Instead, we used steady-state kinetics analysis to investigate the biochemical basis for H_2_O_2_-dependent changes in PKA-C*α*’s activity toward these substrates (Figure 4B,C and Table 4). Interestingly, these data suggest that H_2_O_2_-dependent oxidation differentially affects PKA-C*α*-substrate interactions. For instance, like CREBtide, H_2_O_2_-dependent oxidation of PKA-C*α* alters its affinity for Kemptide. However, unlike CREBtide, which exhibited a decrease in its affinity following pre-treatment with H_2_O_2_, oxidation of PKA-C*α* led to a 2.2-fold *increase* in its affinity for Kemptide (Figure 4B and Table 4). Indeed, untreated PKA-C*α* exhibited a K_m,app_ for Kemptide of 44.0 ± 9.3 µM, which is similar to reported values [36]. However, following H_2_O_2_-dependent oxidation, the K_m,app_ of PKA for Kemptide decreased to 20.1 ± 3.6 µM (Table 4). In contrast, the observed changes in K_m,app_ for Crosstide were much more modest, decreasing by only 15%, from 73.4 ± 3.6 µM for untreated PKA-C*α* to 63.6 ± 17.5 µM for oxidized PKA-Cα (Table 4). Instead, much larger changes in *k_cat,app_* were observed for Crosstide. Indeed, PKA-C*α*’s *k_cat,app_* toward Crosstide increased 3.5-fold following oxidation while its *k_cat,app_* toward Kemptide and GST-CREB only increased ~1.7- and ~2.3-fold, respectively, under the same conditions.

## 4. Discussion

Here, we have investigated the effect of different redox modifications on PKA-Cα substrate selection using the model peptide and protein substrates, Kemptide, Crosstide, CREBtide, and GST-CREB. Interestingly, these studies suggest that different redox modifications lead to differential changes in PKA-Cα substrate selection (Figure 5). For instance, diamide-mediated oxidation of PKA-Cα induces both intra- and intermolecular disulfide bonds that may inhibit PKA-Cα’s catalytic activity. However, our data suggest that the level of inhibition differs depending on the substrate under study. For instance, phosphorylation of Kemptide and CREBtide, both of which contain a canonical PKA consensus phosphorylation motif of [R/K]-[R/K]-X-[S/T]-X, was strongly inhibited following oxidation of PKA-Cα with high concentrations of diamide. In contrast, phosphorylation of Crosstide, which lacks a basic residue at the -3 position relative to the phosphosite (i.e., the P-3 position), was only marginally inhibited by diamide-mediated oxidation of PKA-Cα. This may be because disulfide bond formation induces conformational changes in the enzyme that disrupt interactions with some substrates (e.g., Kemptide and CREBtide) while having little effect on others (e.g., Crosstide). For example, while both of the positively charged residues in the canonical PKA consensus motif form hydrogen bonds with residues on PKA-Cα’s surface, the residues with which they interact are found in distinct regions of the protein. For instance, in the crystal structure of PKA-Cα bound to a peptide composed of residues 5–24 of the endogenous pseudosubstrate inhibitor, PKI (TTYADFIASGRTGRRNAIHD), the residue in the P-3 position (i.e., R18) forms H-bonds with E127 in the hinge region, the backbone carbonyl of T51 in the N-lobe, and the 3′-OH of ATP in the active site cleft (Figure 5A) [37]. In contrast, R19, which is found at the P-2 position, interacts with E170 and E230, both of which are located in the C-lobe. Previously, Humphries et al. demonstrated that diamide-mediated oxidation likely inhibits PKA-Cα activity through the formation of an intramolecular disulfide bond with C343, the only other reactive Cys residue in the enzyme [28]. Since C343 resides in the N-lobe, it is possible that intramolecular disulfide bond formation induces a conformational change that disrupts interactions between residues in the N-lobe/hinge region and basic residues at the P-3 position of the substrate while having little effect on interactions between residues in the C-lobe and positively charged residues at the P-2 position. In this scenario, one of the two primary binding sites for Kemptide and CREBtide (i.e., the Arg at the P-3 position) could potentially be disrupted by diamide-mediated oxidation while maintaining interactions with the basic residues at the P-2 position (which are shared by all three peptides). Meanwhile, Crosstide’s primary site of interaction at the P-2 position would be relatively unperturbed. As a consequence, more dramatic changes in PKA-Cα’s activity toward Kemptide and CREBtide would be expected following diamide-mediated oxidation compared to Crosstide, consistent with our data over the diamide titration series. In support of this hypothesis, if Arg is substituted for Pro in the P-3-position of Crosstide (i.e., Crosstide(P-3→R); GRRRTSSFAEG), the diamide titration curve looks very similar to that observed for Kemptide and CREBtide (Appendix A). This may also explain why subsequent glutathionylation, which largely eliminated the intramolecular disulfide bonding observed following treatment with diamide alone, attenuated and even partially reversed diamide-dependent inhibition of Kemptide and CREBtide (Figure 1). On the other hand, since glutathionylation introduces a bulky moiety near the P + 1 loop substrate binding region, this modification may disrupt or otherwise alter interactions with some substrates while having little effect on others (Figure 5B). In the future, it will be interesting to determine whether similar trends emerge for other PKA substrates with similar characteristics.

Unlike diamide, oxidation of PKA-Cα by the physiological oxidant, H_2_O_2_, did not lead to the formation of either intra- or intermolecular disulfide bonds. Perhaps more surprisingly, H_2_O_2_-dependent oxidation of PKA-Cα increased its activity toward all three substrates tested. This is in stark contrast to the inhibition caused by diamide-mediated oxidation. Peak activity of PKA-Cα toward all three substrates occurred following treatment with 5 µM H_2_O_2_ (corresponding to a 20-fold molar excess of H_2_O_2_ over PKA-Cα) before decreasing at higher peroxide concentrations. Because PKA-Cα oxidized by 5 µM H_2_O_2_ can be glutathionylated by subsequent treatment with GSH-biotin, it is likely that this ratio of H_2_O_2_-to-PKA-Cα leads to sulfenylation of the enzyme (Figure 2C). Sulfenylation would increase the size of the side chain on C199 (essentially adding an oxygen atom) and, as an acid, may also alter the charge state under physiological conditions. These changes could enhance interactions with those substrates that interact better with the sulfenylated residue while disrupting interactions with those that do not (e.g., via steric hindrance or charge repulsion) (Figure 5B). At the same time, H_2_O_2_-dependent oxidation of PKA-Cα could have little to no effect on those substrates that are more readily able to accommodate such a change. This is consistent with the observation that H_2_O_2_-dependent oxidation of PKA-Cα increased its affinity for Kemptide over two-fold while *decreasing* its affinity for CREBtide and GST-CREB and having a negligible effect on its affinity for Crosstide (Figure 3 and Figure 4). In the case of CREBtide, SPR analysis suggests that the change in affinity is likely due to a decrease in *k_off,app_* while having little effect on *k_on,app_*. At the same time, sulfenylation may induce conformational changes and/or promote interactions that enhance its rate of phosphorylation. Consistent with this notion, the apparent catalytic rate constant, *k_cat,app_*, increased toward GST-CREB, Kemptide, and Crosstide following H_2_O_2_-dependent oxidation (however, as with K_m_, the magnitude of these changes differed depending on the substrate under study). Similarly, higher concentrations of H_2_O_2_ may lead to the formation of higher-order oxoforms, such as sulfinic acid and sulfonic acid, which could differentially affect PKA-Cα’s activity toward different substrates. This may account for the differences that we observed in the phosphorylation profiles of each substrate at the higher H_2_O_2_ concentrations. Interestingly, PKA activity has been shown to be modulated by redox-dependent signaling mechanisms during several key cellular processes [38,39,40,41]. For instance, in adipocytes, insulin-induced H_2_O_2_ production leads to decreased rates of lipolysis. This decrease correlated with H_2_O_2_-dependent inhibition of PKA activity, as assessed by Kemptide phosphorylation [38]. In contrast, Srinivasan et al. recently reported that, during hypoxia in both RAW 264.7 macrophages and an in vitro perfused mouse heart system, PKA-mediated phosphorylation of the cytochrome c oxidase (CcO) complex is increased via a ROS-dependent mechanism [41]. While H_2_O_2_-dependent inhibition of PKA in adipocytes is believed to be due to the formation of a disulfide bond between C199 in the catalytic subunit and C97 in the regulatory RII subunits expressed in adipocytes, the PKA catalytic subunit does not form a similar linkage with the RI subunits present in macrophages because C97 is not conserved in RI (though a disulfide may form between the two RI subunits in the type I holoenzymes) [42]. Interestingly, the sites on CcO that are phosphorylated by PKA during hypoxia are different from those modified during normoxia, suggesting that redox modification may even alter site selection within the same substrate [41].

In addition to its downstream substrates, PKA-Cα also recognizes key regulatory factors via redox-sensitive ligand-binding regions. For instance, interactions between PKA-Cα and regulatory factors, such as the endogenous PKA inhibitor, PKI, or the PKA regulatory subunits, RI and RII, are mediated by PKA-C’s P + 1 loop [40,42,43,44]. Therefore, redox modification of PKA-Cα may alter its activation profile inside cells. Consistent with this notion, we have observed a ~2-fold increase in the affinity of PKI (5–24) for oxidized PKA-Cα (Appendix A). Like RI subunits, PKI inhibits PKA-C activity via interactions between its pseudosubstrate domain and PKA-C’s P + 1 loop [40,43,45]. In addition, PKI also promotes PKA-C’s nuclear export, suggesting that redox modification of PKA-C may alter its subcellular localization and its activity toward nuclear targets, such as CREB [46]. Finally, it is interesting to note that the redox-sensitive C199 in PKA-Cα is highly conserved among other AGC family members, suggesting that redox modification may represent a general mechanism of regulation within this kinase family (Appendix A). Consistent with this hypothesis, Byrne and colleagues recently reported that H_2_O_2_-dependent oxidation of several AGC family members, including PKA-Cα, altered their activity toward their cognate substrates [47]. In the case of PKA-Cα, H_2_O_2_-dependent oxidation of the kinase led to a dose-dependent decrease in its activity toward Kemptide. This is in contrast to the dose-dependent increase in PKA-Cα activity toward Kemptide that we observed during our H_2_O_2_ titration assays (Figure 2A). This apparent discrepancy between the two studies may stem from differences in the relative ratios of H_2_O_2_-to-PKA-Cα used in the assays. For instance, at the lowest H_2_O_2_ concentration used by Byrne at al. (i.e., 100 µM), the molar ratio of H_2_O_2_-to-PKA-Cα was 333,333:1 (100 µM H_2_O_2_:0.3 nM PKA-Cα). Meanwhile, we observed a maximal increase in PKA-Cα activity toward Kemptide at a 20:1 molar ratio (5 µM H_2_O_2_:250 nM PKA-Cα). Indeed, at higher H_2_O_2_ concentrations (corresponding to molar ratios between 60:1 and 160:1), we observed a ~15% decrease in PKA-Cα activity toward Kemptide, which is on par with the level of inhibition observed by Byrne et al. using 100 µM H_2_O_2_. This may suggest that PKA-Cα is initially sulfenylated at relatively low H_2_O_2_ concentrations, leading to an increase in activity toward Kemptide (and CREBtide and Crosstide in our assays) before undergoing hyperoxidation to sulfinic and/or sulfonic acid that may decrease its activity toward these substrates. At very high H_2_O_2_-to-PKA-Cα ratios (e.g., an ~3.3 × 10^7^:1 ratio where Byrne et al. observed an ~75% decrease in PKA-Cα activity toward Kemptide), oxidation of C343 may promote intramolecular disulfide bond formation similar to what was observed following diamide treatment, inhibiting PKA-Cα’s activity toward Kemptide to a greater extent. Interestingly, even at the highest H_2_O_2_ concentrations used in our assays (corresponding to a 160:1 molar ratio), PKA-Cα-mediated phosphorylation of CREBtide remained elevated relative to the untreated control while its activity toward both Kemptide and Crosstide fell below baseline, underscoring the differences observed between substrates. Together, these data suggest that redox modification of PKA-Cα differentially alters its activity toward different substrates. In the future, it will be interesting to explore the molecular mechanisms underlying oxidation-induced changes in PKA-Cα-substrate interactions.

## Figures and Tables

**Figure 1 life-13-01811-f001:**
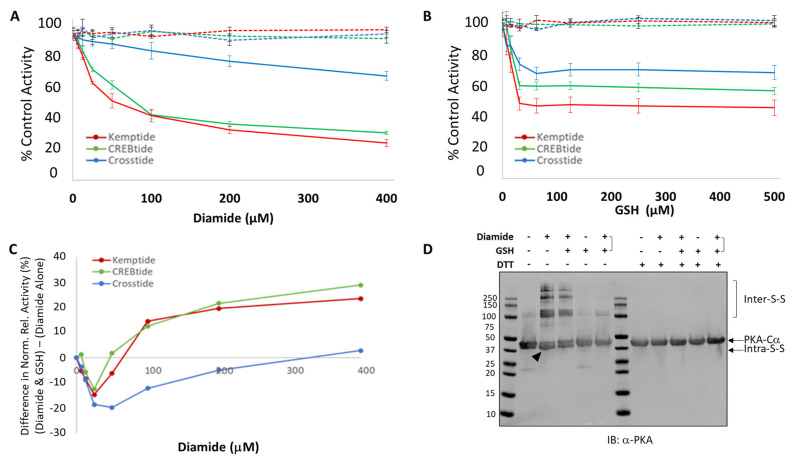
Impact of diamide-mediated oxidation and subsequent glutathionylation on PKA-Cα activity toward different model substrates. (**A**). Normalized relative activity of PKA-Cα toward Kemptide (red), CREBtide (green) and Crosstide (blue) following pre-treatment with the indicated concentrations of diamide for 10 min at room temperature. Wild-type curves are shown as solid lines while PKA-Cα (C199S) curves are shown as dashed lines. (**B**). PKA-Cα activity toward Kemptide, CREBtide, and Crosstide after pre-treatment with diamide for 10 min at room temperature followed immediately by incubation with a 1.25-fold molar excess of reduced glutathione (GSH) for an additional 10 min at room temperature. (**C**). Difference in the normalized relative activity of PKA-Cα treated with diamide followed by GSH as in (**B**) (diamide and GSH) and PKA-Cα treated with diamide alone as in (**A**) (diamide alone). (**D**). Purified PKA-Cα was incubated in the presence of 100 μM diamide alone (lanes 3 and 9), 100 μM diamide followed by 125 μM GSH (lanes 4 and 10), 125 μM GSH alone (lanes 5 and 11), or 100 µM diamide and 125 μM GSH simultaneously (lanes 6 and 12). Samples were then resolved by SDS-PAGE in the presence (right) or absence (left) of dithiothreitol (DTT) reducing agent and analyzed by western blotting using an anti-PKA-Cα antibody. (**A**) black arrowhead indicates the position of a faster migrating species caused by the formation of an intramolecular disulfide bond (Intra-S-S) while brackets to the right indicate a series of slower migrating species caused by the formation of intermolecular disulfide bonds (Inter-S-S).

**Figure 2 life-13-01811-f002:**
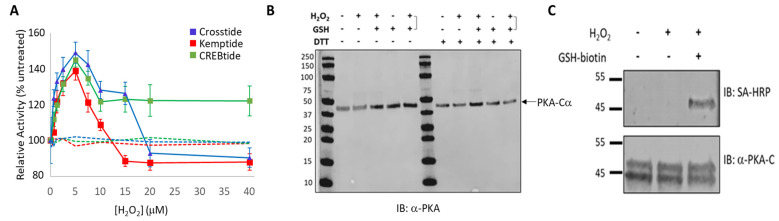
Impact of H_2_O_2_-dependent oxidation of PKA on substrate selection. (**A**). Effect of H_2_O_2_-dependent oxidation on PKA-Cα activity toward Crosstide (blue), Kemptide (red), and CREBtide (green). Wild-type curves are shown as solid lines while PKA-Cα (C199S) curves are shown as dashed lines. (**B**). Non-reducing PAGE following treatment of PKA-Cα with 5 µM H_2_O_2_ alone (lanes 3 and 9), 5 µM H_2_O_2_ followed by 25 µM reduced glutathione (GSH) (lanes 4 and 10), GSH alone (lanes 5 and 11), or 5 µM H_2_O_2_ and 25 µM GSH at the same time. Samples were then resolved by SDS-PAGE in the presence (right) or absence (left) of dithiothreitol (DTT) reducing agent and analyzed by western blot using an anti-PKA-Cα antibody. (**C**). Western blot following treatment with dH_2_O (lane 1), 5 µM H_2_O_2_ alone (lane 2), or 5 µM H_2_O_2_ followed by 25 µM GSH-biotin (lane 3). The blot was then probed with streptavidin-HRP (SA-HRP) followed by mouse anti-PKA-Cα antibody (α-PKA-C).

**Figure 3 life-13-01811-f003:**
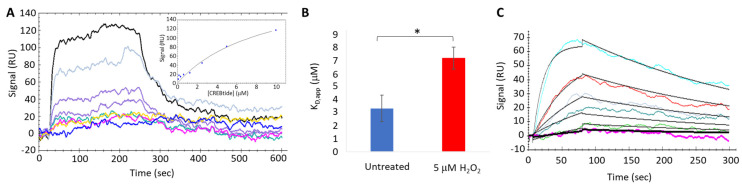
Effects of H_2_O_2_-dependent oxidation on PKA-C*α*-CREBtide interactions. (**A**). Representative equilibrium binding experiment measuring the affinity of oxidized PKA-Cα for CREBtide using surface plasmon resonance imaging (SPRi) at PKA-Cα concentrations ranging from 0.156–10 µM. Different colored lines represent PKA-Cα concentrations ranging from 10 μM (black) to 0.156 μM (royal blue). The binding curve is shown in the inset. (**B**). Average K_D,app_ for untreated PKA-Cα (blue) and PKA-Cα treated with 5.0 μM H_2_O_2_ before injection (red). Error bars represent standard error about the mean (n = 4). Statistically significant differences are indicated by an asterisk * (*p* < 0.05). (**C**). Representative kinetic binding experiment measuring the on- and off-rates (*k_on_* and *k_off_*, respectively) of oxidized PKA-Cα for CREBtide using SPRi. Different collored lines represent PKA-Cα concentrations ranging from 10 μM (light blue) to 0.156 μM (magenta).

**Figure 4 life-13-01811-f004:**
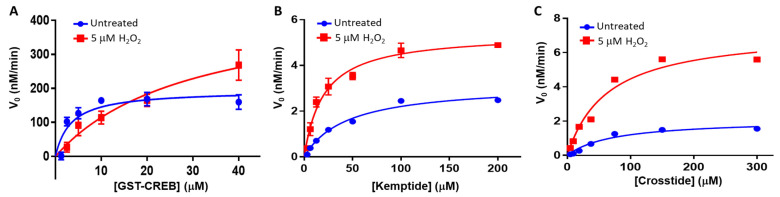
Steady-state kinetics analysis of H_2_O_2_-dependent changes in PKA-C*α*-substrate interactions. PKA-Cα (250 nM) was pre-treated with either dH_2_O (blue; untreated) or 5 µM H_2_O_2_ (red; 5 µM H_2_O_2_) for 10 min before excess H_2_O_2_ was scavenged with catalase for 1 min. The treated kinase was then incubated with the indicated concentrations of GST-CREB (**A**), Kemptide (**B**), or Crosstide (**C**) for 30 min at 30 °C before measuring kinase activity using the ADP-Glo assay (Promega). Error bars represent standard error about the mean of at least three independent experiments conducted in duplicate.

**Figure 5 life-13-01811-f005:**
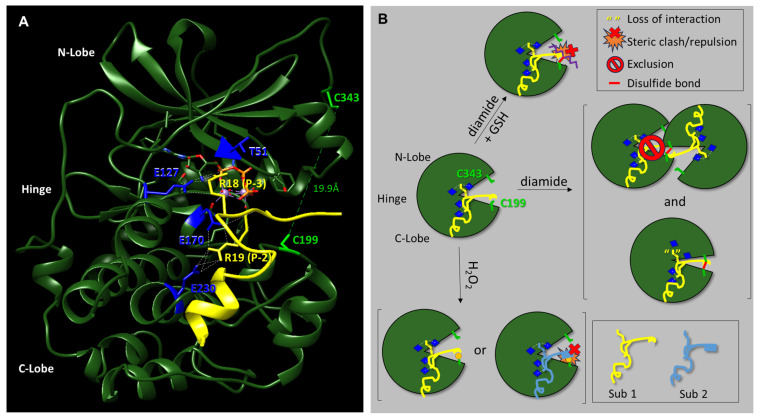
Models of redox-dependent changes in PKA-Cα-substrate interactions. (**A**). Co-crystal structure of murine PKA-Cα (green) bound to the inhibitory peptide, PKI (5–24) (yellow). PKA-Cα residues E127 and T51 (blue sticks) in the hinge region and N-lobe, respectively, interact with R18 in the P-3 position of PKI (5–24) (yellow sticks) while PKA-Cα residues E170 and E230 (blue sticks) in the C-lobe interact with R19 in the P-2 position of PKI (5–24) (yellow sticks). The positions of C199 and C343, the only two Cys residues present in PKA-Cα, are shown as green sticks. The structure was generated from PDB ID: 1ATP using ChimeraX [34,37]. (**B**)**.** Cartoon models depicting how diamide- and H_2_O_2_-dependent oxidation of PKA-Cα, as well as subsequent glutathionylation with reduced glutathione (GSH), may differentially affect its interactions with different substrates. Residues involved in interactions with basic residues in the substrates (i.e., E127 and T51 in the hinge region and N-lobe, respectively, and E170 and E230 in the C-lobe) are depicted as blue diamonds while C199 and C343 are depicted as green sticks. Diamide-dependent oxidation leads to the formation of either intermolecular disulfide bonds between two or more PKA-Cα subunits or an intramolecular disulfide bond between C199 and C343 in the same molecule. Intermolecular disulfide bond formation may block access of some substrate molecules to the active site (i.e., “exclusion”). Meanwhile, rotation of the N-lobe about the hinge region to promote intramolecular disulfide bond formation may displace residues in the hinge region/N-lobe involved in substrate binding (e.g., E127 and T51) while having little effect on the position of residues in the C-lobe (e.g., E170 and E230). Displacement of residues in the hinge region/N-lobe may cause some PKA-Cα-substrate interactions to be lost (yellow quotation marks) while those with residues in the C-lobe remain unaffected. Subsequent glutathionylation forms a mixed disulfide that may cause steric clashes with some substrates (orange starburst). In the case of H_2_O_2_, redox modification (e.g., sulfenylation) may promote interactions with some substrates (e.g., yellow Sub 1) through the introduction of additional binding sites while preventing interactions with other substates (e.g., light blue Sub 2) due to steric clashes and/or charge repulsion.

**Table 1 life-13-01811-t001:** Primer sets used to construct PKA (C199A) and PKA (C199S) by QuikChange mutagenesis.

Primer	Sequence (5′ -> 3′)	Length	T_m_ (°C)
PKA (C199A)_F	GCT GGG ACC CCT GAG TAC TTG GCC CCC GAG ATT ATC	36	69.3
PKA (C199S)_F	TCT GGG ACC CCT GAG TAC TTG GCC CCC GAG ATT ATC	36	68.7
PKA (C199A or S)_R	CAA GGT CCA AGT ACG GCC TTT CAC ACG CTT GGC A	34	68.4

**Table 2 life-13-01811-t002:** QuikChange mutagenesis conditions.

Step	Temperature (°C)	Duration
1	98	1 min
2	98	15 s
3	70	15 s
4	72	4 min
5	Go to 2	25×
6	72	7 min
7	10	forever

**Table 3 life-13-01811-t003:** Kinetic binding analysis of PKA-CREBtide using SPR.

Condition	*k_on,app_* (M^−1^s^−1^) (×10^3^)	*k_off,app_* (s^−1^) (×10^−2^)	K_D,app_ (μM)
Untreated	6.71 ± 1.21	1.63 ± 0.25	2.80 ± 0.75
5 μM H_2_O_2_	8.07 ± 1.65	4.77 ± 0.69	6.22 ± 0.70
Fold change	1.20-f	2.93-f	2.22-f
*p*-value	0.532	0.015	0.013

**Table 4 life-13-01811-t004:** Steady-state kinetics analysis of PKA.

Substrate	Condition	*k_cat,app_* (s^−1^)	K_m,app_ (μM)	k_cat_/K_m_ (μM^−1^s^−1^)
GST-CREB	untreated	12.9 ± 2.1	3.39 ± 1.9	3.8
	5 μM H_2_O_2_	30.4 ± 6.9	29.8 ± 12.4	1.0
Kemptide	untreated	210.5 ± 16.5	44.0 ± 9.3	4.8
	5 μM H_2_O_2_	357.7 ± 19.3	20.1 ± 3.6	17.8
Crosstide	untreated	138.3 ± 17.9	73.4 ± 24.7	1.9
	5 μM H_2_O_2_	483.7 ± 29.1	63.6 ± 17.5	7.6

## Data Availability

Data are contained within the article or Appendix A.

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
