# Peer review of "Redox Modification of PKA-Cα Differentially Affects Its Substrate Selection"

_life, 2023, doi:10.3390/life13091811_

Round 1

Reviewer 1 Report

In this manuscript, Delva-Wiley and colleagues examine the effect of PKA oxidation/glutathionylation on in vitro substrate selectivity and binding affinity. Both modifications on PKA-C199 has been reported previously and here the authors present their key finding that C199 modification may direct PKA to particular substrates. The finding is likely to stimulate further research and will be of interest to many working on kinases that harbor an analogous Cys residue at this position. The study is quite preliminary, based mainly on peptide substrates using ADP-Glo assays and SPR binding analysis. However, experiments are well-performed and conclusions drawn are largely consistent with the data as described. Overall, this is a solid biochemical/biophysical study that poses significant questions concerning fundamental regulation of a major metabolic signalling network.

Major comments:

1. Line 504-506: Diamide with subsequent GSH incubation is stated as eliminating PKA-Ca intramolecular disulfide bonds and decreased the extent of intermolecular bonding but this is not at all apparent from Fig 1D (I am comparing lanes 3 & 4). In particular, I could not see which band(s) the Intra-S-S arrow is referring to. The authors should provide some quantitation to back up both statements. This is important because it forms the basis for data interpretation on line 613.

2. In the discussion the authors speculate on structural aspects explaining their results. Why not perform similar experiments described in this study to support these ideas? For example it would be relatively simple to analyse the effect of diamide treatment on phosphorylation of a custom synthesised Kemptide-like peptide without a basic residue at P-3, or conversely Crosstide with a P-3 basic residue.

3. Line 601: while C343 may be in the PKA C-lobe, structures reveal it is located near the B-helix in the N-lobe and quite distant from the P-2/P-3 binding regions. How does this fit with the speculation here that diamide treatment only disrupts interactions with the P-3 residue?  

Minor comments:

4. Lines 313 & 332: should ml be ul? Similarly lines 320 & 343 – are peptide substrate concs mM or uM?

5. Fig 3A/C: For immediate understanding please indicate which SPR traces are high H2O2, and which are low.

6. Fig 1A/B How do basal levels of PKA activity compare across the 3 peptide substrates?

7. Line 684/689: I suggest replacing “their data” with an unequivocal term.

8. p14: Table 4 seems to be listed twice as Table 2 in the text.

9. Fig S3: I believe PRKAA2 C174 has also been shown to be oxidised.

http://dx.doi.org/10.1016/j.cmet.2013.12.013

Reviewer 2 Report

1.       Page 1, line 12: “…subunit of PKA (PKA-Cα) is oxidized on C199,…” Authors should indicate which protein they refer to in the text, not necessarily in the abstract but definitely somewhere in the beginning of the manurscript. The proper way is to give a Uniprot number.

2.       In figure 1S authors refer to PDB: 1ATP this structure corresponds to Uniprot ID:  P05132 (KAPCA_MOUSE). The particular protein (https://www.uniprot.org/uniprotkb/P05132/entry) has a C residue at position 200 and not 199 that is referred though the text. Please clarify

3.       Page 10, line 461-465: “…Previously, Humphries et al. demonstrated that PKA-CαC199 is oxidized by diamide and that this modification decreased its activity toward the model peptide substrate, Kemptide [29]. To determine ……three model PKA substrates, namely Kemptide, CREBtide, and Crosstide. While Kemptide (LRRASG)..”

In reference 29 Kenneth M. Humphries, et al (EXPERIMENTAL PROCEDURES page 43506) they refere  to Kemptide as  Leu-Arg-Arg-Ala-Ser-Leu-Gly this is not the same sequence as the one that authors used although they use the above reference. Please clarify

4.        Page 7, lines 309-310:Where indicated, diamide-treated samples were subsequently incubated with 0-2 mM GSH for 10 minutes at room temperature” If samples were subsequently incubated with GSH it means at the end of the experiment they will be treated 20min in diamide and 10 min in that will make incomparable the results in figure 1A and 1B.

5.       Page 11 figure 1C. “Difference in normalized relative activity between diamide & GSH and diamide alone” such a sort sentence is not enough to explain such complicate figure please expand. For example, is the Y axes difference in activity? If yes. why the Y axes is referred as “difference” and not “difference in activity”

6.       Page 11, figure 1. is there any evidence or any indication how incubation time may affect PKA Ca activity?

Diamide & GSH concentration used in order to see an effect were high (30-100µM) compere to enzyme (0,3 µM). So it is very important to know if the time is the crucial factor rather than the concentration. In other words, how PKA-Ca activity is effected if you incubate the enzyme in low concentration of Diamide (e.g 30 µM) for 1 or 3 h?

7.       Page 14: Figure 4. Steady-state kinetics analysis of H2O2-dependent changes in PKA-Cα-substrate interactions. PKA-Cα(250 nM) was pre-treated with either dH2O (blue; untreated) or 5 µM H2O2(red; 5 mM H2O2) for 10 minutes before excess…”

It is not clear if the “red curves” are corresponded to 5 µM H2O2 or 5 mM H2O2 please clarify.

Reviewer 3 Report

Delva-Wiley et al. report in this manuscript the resulta of an accurate investigation on the effects of redox modifications of  cyclic AMP-dependent protein kinase (PKA) , focusing on its C-alpha catalytic subunit and its highly conserved C199 cysteine residue. Diamide and hydrogen peroxide were used as prooxidants, and their differential effects on PKA-C-alpha substrate specificity were compared with respect to selected model peptides, using biochemical as well as biophysical analytical techniques. The data obtained show that diamide oxidation has a general inhibitory effect, while at variance a biphasic effect of hydrogen peroxide is apparent, with a stimulation at low concns. vs. an inhibition at higher ones.. As stated by the authors, their paper is  offering novel insights into the crosstalk between redox- and phosphorylation-dependent signaling pathways mediated by PKA. The manuscript is well written, clear and interesting, The illustrations are accurate and informative, the references list is updated and comprehensive. As such the paper indeed represents a good piece of science, forming the basis for subsequent studies aiming at elucidating the underlying molecular mechanisms.

Specific comments.

1. details of the SPRi analysis should be briefly introduced, in order to make its rationale accessible to non-expert readers as well;

2. the authors may want to insert 1-2 cartoons summarizing the differential results obtained with the two prooxidants, also including the envisaged biochemical mechanisms (sulfenylation, glutathionylation, disulfide bond formation…) and the alleged changes in interactions of PKA with its inhibitor PHI.

Round 2

Reviewer 1 Report

The authors have addressed all of my comments.